# A Systematic Comparison of Antiandrogens Identifies Androgen Receptor Protein Stability as an Indicator for Treatment Response

**DOI:** 10.3390/life11090874

**Published:** 2021-08-25

**Authors:** Tiziana Siciliano, Ingo H. Simons, Alicia-Marie K. Beier, Celina Ebersbach, Cem Aksoy, Robert I. Seed, Matthias B. Stope, Christian Thomas, Holger H. H. Erb

**Affiliations:** 1Department of Urology, Technische Universität Dresden, 01307 Dresden, Germany; tiziana.siciliano@uniklinikum-dresden.de (T.S.); ingos@mailbox.org (I.H.S.); aliciamarie.beier@uniklinikum-dresden.de (A.-M.K.B.); celina.ebersbach@uniklinikum-dresden.de (C.E.); cem.aksoy@uniklinikum-dresden.de (C.A.); christian.thomas@uniklinikum-dresden.de (C.T.); 2Mildred Scheel Early Career Center, Department of Urology, Medical Faculty, University Hospital Carl Gustav Carus, 01307 Dresden, Germany; 3Department of Pathology, University of California, San Francisco, CA 94110, USA; Robert.Seed@ucsf.edu; 4Department of Gynecology and Gynecological Oncology, University Hospital Bonn, 53127 Bonn, Germany; Matthias.Stope@ukbonn.de; 5UroFors Consortium (Natural Scientists in Urological Research) of the German Society of Urology, 14163 Berlin, Germany

**Keywords:** prostate cancer, AR signaling, nuclear receptor, therapy resistance, proteasomes

## Abstract

Antiandrogen therapy is a primary treatment for patients with metastasized prostate cancer. Whilst the biologic mechanisms of antiandrogens have been extensively studied, the operating protocols used for the characterization of these drugs were not identical, limiting their comparison. Here, the antiandrogens Bicalutamide, Enzalutamide, Apalutamide, and Darolutamide were systematically compared using identical experimental setups. Androgen-dependent LNCaP and LAPC4 cells as well as androgen-independent C4-2 cells were treated with distinct concentrations of antiandrogens. Androgen receptor (AR)-mediated gene transactivation was determined using qPCR. Cell viability was measured by WST1 assay. Protein stability and AR localization were determined using western blot. Response to the tested antiandrogens across cellular backgrounds differed primarily in AR-mediated gene transactivation and cell viability. Antiandrogen treatment in LNCaP and LAPC4 cells resulted in AR protein level reduction, whereas in C4-2 cells marginal decreased AR protein was observed after treatment. In addition, AR downregulation was already detectable after 4 h, whereas reduced AR-mediated gene transactivation was not observed before 6 h. None of the tested antiandrogens displayed an advantage on the tested parameters within one cell line as opposed to the cellular background, which seems to be the primary influence on antiandrogen efficacy. Moreover, the results revealed a prominent role in AR protein stability. It is one of the first events triggered by antiandrogens and correlated with antiandrogen efficiency. Therefore, AR stability may surrogate antiandrogen response and may be a possible target to reverse antiandrogen resistance.

## 1. Introduction

The androgen receptor (AR) is the main driving force behind the growth and progression of prostate cancer (PC), the most common carcinoma in men [1,2]. The AR belongs to the nuclear receptor family and mediates the actions of androgens. Its topology includes four functional domains: the N-terminal transactivation domain, the Deoxyribonucleic acid (DNA)-binding domain, the hinge region, and the ligand-binding domain. AR signaling is controlled by androgen binding. Consequently, androgen withdrawal causes reduced cell growth and induction of apoptosis [2,3]. Hence, androgen axis targeting became the standard systemic therapy in PC [4,5]. In metastatic PC, the AR signaling blockade is achieved by administering antiandrogens, which compete with endogenous androgens for AR binding and inhibit AR-mediated gene transactivation [5]. According to the European Association of Urology (EAU) guidelines, the antiandrogens most commonly used in clinic-based treatments are Enzalutamide (Xtandi^®^), Apalutamide (Erleada^®^), and Darolutamide (Nubeqa^®^) [6,7]. First-generation antiandrogens such as Bicalutamide (Casodex^®^), Flutamide (Eulexin^®^), or Nilutamide (Nilandron@) are still available but are solely used for the treatment of the androgen flare-up phenomena, a common effect of the luteinizing hormone-releasing hormone (LHRH) agonist treatment [6,7]. All antiandrogens can either be used independently or combined with surgical or chemical castration to obtain maximum androgen blockade.

In the last 3 decades, several cell line models have been introduced to PC research. These include androgen-dependent cell lines (e.g., LNCaP, LAPC4, MDA PCa 2B) and androgen-independent cell lines (e.g., C4-2, PC3, Du145) [8]. In addition, several 2D and 3D primary human and mouse models have been developed [8,9,10]. These novel 2D and 3D models are closer to PC biology in patients than the established cell lines. However, cell line models provide high genetic stability and low heterogeneity and are therefore ideally suited for research into signaling pathways [11].

The effects of antiandrogens on AR-signaling have been investigated extensively. However, a direct comparison of these drugs on AR-signaling is limited. The experimental procedures used for their initial characterization differ in used cell models, cellular assays, culture conditions, incubation times, and drug concentrations. For example, Darolutamide was tested with AR-HEK293 cells, and Bicalutamide was tested with LNCaP/AR cells [12,13]. Furthermore, Darolutamide was tested with WST-1 cell viability assay and Bicalutamide with CyQUANT cell proliferation assays [12,14]. In addition, for Bicalutamide testing, RPMI 1640 medium was used, whereas for Enzalutamide testing, the IMDM medium was utilized [13,15]. Moreover, the drug concentration varies between 0–10 µM in Enzalutamide and 0–1 µM in Bicalutamide [14,16]. The incubation time of the drugs differs from 48 h in Apalutamide and 1–4 days in Enzalutamide [13,16]. Therefore, this study aimed to systematically compare the current EAU recommended antiandrogens Bicalutamide, Enzalutamide, Apalutamide, and Darolutamide, using identical experimental setups and drug concentrations.

## 2. Materials and Methods

### 2.1. Cell Culture

The human PC cell lines LNCaP and PC3 were obtained from the American Type Culture Collection (ATCC). C4-2 cells were kindly provided by Prof. Thalmann (University of Berne, Berne, Switzerland) [17]. Dr A. Cato (University of Karlsruhe, Karlsruhe, Germany) provided the cell line LAPC4. PC3, LAPC4, and LNCaP were cultured as described previously [18]. Characteristics of the cell lines are displayed in Table 1. Mycoplasma testing was routinely performed using the Mycoalert Detection Assay (Lonza). Cell line authentication was performed yearly by STR profiling.

### 2.2. Drug Treatment

Methyltrienolone (R1881) (Sigma-Aldrich, R0908-10MG, Lot Number: 085M4610V), Bicalutamide (Selleck Chemicals LLC, Tokyo, Japan, S1190, Lot Number: 5), Enzalutamide (Astellas Pharma, Tokyo, Japan, 3343, Lot Number: RS-8BK0189-4), Apalutamide (Selleck Chemicals LLC, Tokyo, Japan, S2840, Lot Number: 2), and Darolutamide (Selleck Chemicals LLC, S75559, Tokyo, Japan, Lot Number: 1) were dissolved in DMSO as a 100 mM stock solution and stored in aliquots at −80 °C. As R1881 is not metabolized as rapidly as the natural 5α-dihydrotestosterone, it was used as androgen for the studies [23]. Competition Binding studies have revealed an IC50 of 160 nM for Bicalutamide and 21.4 nM for Enzalutamide in LNCaP cells [16]. In the presence of 0.45 nM of testosterone, AR luciferase reporter gene assays revealed for Enzalutamide an IC50 of 26 nM, for Apalutamide an IC50 of 200 nM, and for Darolutamide an IC50 of 26 nM [12]. In line with these IC50 values and previous studies, an antiandrogen concentration range of 0.01 µM to 10 µM was chosen [24,25].

24 h after cell seeding, the medium was changed to 5% dextran-coated charcoal treated FCS (FCSdcc; Thermo Fisher Scientific, Waltham, MA, USA) for 24 h to deplete the medium of steroid hormones and growth factors. Subsequently, cells were treated with R1881 alone or with a combination of R1881 and antiandrogens. Treatment concentration and duration were chosen due to previous studies revealing a maximal response after 1 nM R1881 treatment at 16 h [26,27].

### 2.3. Subcellular Fractionation

For subcellular fractionation, cells were washed with ice-cold PBS and directly harvested in 300 µL cytoplasmic lysis buffer (10 mM HEPES pH 7.9, 10 mM KCl, 1.5 mM MgCl_2_, 340 mM Sucrose, 10% Glycerol, 1 mM DTT, protease inhibitor) before the addition of 0.1% Triton X-100 and incubation for 5 min on ice. Subsequently, the suspension was centrifuged at 1300× *g* for 4 min at 4 °C to separate the nuclei from the cytoplasmic proteins. After transferring the cytoplasmic fraction into a new reaction tube, the nuclei pellet was washed once in 500 µL cytoplasmic lysis buffer. After repeated centrifugation, the nuclei were lysed using 100 µL RIPA lysis buffer. According to the manufacturer’s protocol, protein concentrations were determined using the Quick Start™ Bradford Protein Assay (Bio-Rad, Hercules, CA, USA). After separation, NuPAGE™ LDS Sample Buffer (4×) was added to the fractions, and the nuclei samples were sonicated. Then, 20 µg of the fractions was used for western blot analysis. Lamin A/C (nuclear fraction) and GAPDH (cytoplasmic fraction) detection controlled fraction quality (Appendix A).

### 2.4. Western Blot Analysis

Cell harvest, cell lysis, protein determination, and western blot were performed as previously described [18]. 20 µg protein lysate electrophoresis were separated by SDS-gel-electrophorese using NuPAGE™ 4–12% Bis-Tris protein gels and subsequently transferred to a nitrocellulose membrane using the iBlot Dry Blotting System (all Thermo Fisher Scientific, Waltham, MA, USA). 5 µL Spectra Multicolour Broad Range (Thermo Fisher Scientific, Waltham, MA, USA) protein standard and 1 µL MagicMark™ XP Western Protein Standard (Thermo Fisher Scientific, Waltham, MA, USA) were used. For detection, the membranes were incubated with WesternBright Sirius HRP substrate (Advansta, San Jose, CA, USA), and all signals were detected by a Microchemi chemiluminescence system (DNR Bio-Imaging Systems, Jerusalem, Israel). Densitometric analysis of experiments was performed with the Image-Studio Lite 5.2 software (LI-COR, Lincoln, Dearborn, MI, USA). Used antibodies are listed in Table 2. Uncropped western blot images are displayed in the Appendix A.

### 2.5. Ribonucleic Acid (RNA) Isolation and Quantitative Real-Time PCR (qPCR)

Total RNA was isolated using the DIRECT-ZOL RNA MINIPREP (Zymo Research, Freiburg, Germany) following the manufacturer’s instructions. Complementary DNA (cDNA) synthesis was performed with 500 ng total RNA using the Superscript II RNase H Reverse Transcriptase kit (Thermo Fisher Scientific, Waltham, MA, USA). The quantitative real-time polymerase chain reaction (qPCR) was performed with GoTaq Probe qPCR Master Mix (Promega, Mannheim, Germany) and appropriate primers for 45 cycles on a LightCycler 480 instrument (Roche Diagnostics, Mannheim, Germany). The geometric mean of *HPRT1* and *TBP* was used for normalization. ΔCp = Cp_GOI_ − Cp_Housekeeper_ values were calculated and expressed as 2^−ΔCp^. Following primer assays have been used (all Thermo Fisher Scientific, Waltham, MA, USA): *AR* (Hs00171172_m1), *PSA/KLK3* (Hs02576345_m1), *TMPRSS2* (Hs01122322_m1), *PROSTEIN**/SLC45A3* (Hs01026319_g1), *HPRT1* (Hs02800695_m1), *TBP* (Hs00427620_m1).

### 2.6. Measurement of Cell Viability

Cell viability was assessed using the WST-1 cell proliferation reagent (Roche Diagnostics, Mannheim, Germany). Cells (10,000 cells in 100 μL) were seeded into 96-well culture plates. After 24 h, the medium was changed to a 50 µL medium containing 5% FCSdcc (Thermo Fisher Scientific, Waltham, MA, USA) for 24 h. Subsequently, cells were treated with the synthetic androgen R1881 or with a combination of R1881 and antiandrogens diluted in 50 µL Medium containing 5% FCSdcc. After 72 h treatment, 10 µL WST-1 solution was added for an additional 2 h. Subsequently, absorbance was recorded at 450 nm (Reference 620 nm) for each well using Mithras LB 940 (Berthold Technologies, Bad Wildbad, Germany). After subtracting background absorbance, results were calculated as x-fold of 1 nM R1881 treated cells.

### 2.7. Statistical Analysis

Prism 9.1.2 (GraphPad Software, San Dieg, CA, USA) was used for statistical analyses. Differences between treatment groups were analyzed using Two-Way ANOVA with Dunnett’s correction for multiple comparisons. *p*-values of ≤ 0.05 were considered statistically significant. All differences highlighted by asterisks were statistically significant as encoded in figure legends (*: *p* ≤ 0.05; **: *p* ≤ 0.01; ***: *p* ≤ 0.001). All experiments have been performed in at least three biological replicates unless noted otherwise.

## 3. Results

### 3.1. Antiandrogens Affect AR-Mediated Gene Transactivation in a Cell-Specific Manner

To determine the effects of antiandrogens on AR-mediated gene transactivation, the mRNA levels of three AR target genes (*PSA*, *TMPRSS2*, *PROSTEIN*) were detected by qPCR and displayed as a mean change of 1 nM R1881 treated control [28,29,30].

In LNCaP cells, Bicalutamide, Apalutamide, and Darolutamide treatments significantly decreased *PSA* mRNA to ~0.35 fold of 1 nM R1881 at 10 µM (Figure 1A). Only Enzalutamide treatment revealed a reduction of *PSA* mRNA at 0.1 µM and 1 µM. Complete steroid hormone withdrawal reduced *PSA* mRNA down to ~0.30 fold of 1 nM R1881. *TMPRSS2* mRNA is significantly downregulated by Bicalutamide and Enzalutamide treatments to ~0.15 fold of 1 nM R1881 at 10 µM (Figure 1B), whereas Darolutamide only reduced *TMPRSS2* mRNA levels to ~0.35 fold. Moreover, Bicalutamide and Enzalutamide already reduced *TMPRSS2* mRNA levels at 0.1 µM, whereas Darolutamide only showed effects at 10 µM. Complete steroid hormone withdrawal led to a reduction of *TMPRSS2* mRNA down to ~0.10 fold of 1 nM R1881. All antiandrogens used in this study reduce *PROSTEIN* mRNA levels compared to the 1 nM R1881 control (Figure 1C). While Bicalutamide and Enzalutamide reduce *PROSTEIN* mRNA levels down to ~0.10 fold, Apalutamide and Darolutamide reduce *PROSTEIN* mRNA levels only down to ~0.40 fold. Complete steroid hormone withdrawal reduced *PROSTEIN* mRNA down to ~0.15 fold of 1 nM R1881.

In the androgen-independent LNCaP sub-cell line C4-2, all tested antiandrogens reduced *PSA* mRNA only down to ~0.45 fold of 1 nM R1881 at 10 µM (Figure 1D) and complete steroid hormone withdrawal reduced *PSA* mRNA down to ~0.25 fold. In line with this observation, all tested antiandrogens reduced *TMPRSS2* mRNA to ~0.50 fold of 1 nM R1881 at 10 µM (Figure 1E) and complete steroid hormone withdrawal reduced *TMPRSS2* mRNA to ~0.25 fold. Bicalutamide and Enzalutamide reduced *PROSTEIN* mRNA levels to ~0.30 fold at 10 µM, whereas Apalutamide and Darolutamide reduced *PROSTEIN* mRNA levels only to ~0.45 fold. Complete steroid hormone withdrawal reduced *PROSTEIN* mRNA to ~0.25 fold of 1 nM R1881.

Antiandrogen treatment of the cell line LAPC4 resulted in downregulation of *PSA* mRNA to ~0.45 fold of 1 nM R1881 at 1 µM and ~0.05 fold at 10 µM. Darolutamide effectively reduced *PSA* mRNA levels down to 0.30 fold at 0.1 µM (Figure 1G). Complete steroid hormone withdrawal led to a reduction of *PSA* mRNA down to ~0.05 fold of 1 nM R1881. Bicalutamide and Enzalutamide treatment reduced *TMPRSS2* mRNA levels down to ~0.30 fold of 1 nM R1881 at 1 µM and down to ~0.10 fold at 10 µM, whereas Apalutamide and Darolutamide showed only a reduction of *TMPRSS2* mRNA down to ~0.50 fold at 10 µM (Figure 1H). Complete steroid hormone withdrawal led to a decrease of *TMPRSS2* mRNA down to ~0.45 fold of 1 nM R1881. Regulation of *PROSTEIN* mRNA levels by antiandrogen treatment in LAPC4 cells follow a similar pattern as *TMPRSS2* mRNA. Bicalutamide and Enzalutamide reduce *PROSTEIN* mRNA down to ~0.10 fold at 10 µM, Apalutamide and Darolutamide down to ~0.40 fold at 10 µM, and complete steroid hormone withdrawal led to a reduction of *PROSTEIN* mRNA down to ~0.35 fold of 1 nM R1881 (Figure 1I).

These cell line comparisons revealed that C4-2 cells were affected with the lowest and LAPC4 cells with the highest sensitivity to antiandrogen treatment (Appendix A).

### 3.2. Cell Viability Is Affected in a Cell-Specific Manner by Antiandrogens

In LNCaP cells, only Enzalutamide treatments resulted in a concentration-dependent decrease in cell viability (~0.55 fold) which was similar to the level of complete steroid hormone withdrawal (~0.53 fold of 1 nM R1881), starting at a concentration of 1 µM and peaking at 10 µM (Figure 2A). The other antiandrogens only inhibited cell viability at 10 µM in LNCaP cells down to ~0.70 fold.

In C4-2 cells, Bicalutamide treatment resulted in a significant decrease in cell viability at 10 µM to ~0.70 fold (Figure 2B). Therefore, only Bicalutamide reduced cell viability to a level similar to complete steroid hormone withdrawal (~0.75 fold of 1 nM R1881).

Similar to the results seen in LAPC4 cells on AR-activity, all anti-androgen treatments only caused a significant reduction in cell viability (~0.50 fold) when used at 10 µM. Except for Enzalutamide, a decrease in cell viability starts at a concentration of 0.1 µM and peaks at 10 µM (Figure 2C). Only Apalutamide already peaked at a concentration of 1 µM.

Comparison of the influence of the tested antiandrogens on cell viability reveals that the cell line C4-2 responded with the lowest and LAPC4 with the highest sensitivity to antiandrogen treatment (Figure 2D–G).

### 3.3. Bicalutamide, Enzalutamide, Apalutamide, and Darolutamide Reduce Nuclear AR Protein Levels after Treatment

To assess the impact of antiandrogens on AR localization, LNCaP, C4-2, and LAPC4 cells were starved for 24 h followed by non-treatment, 1 nM R1881 treatment, and different concentrations of antiandrogens for 2 h. Subsequently, cells were fractionated, and AR levels were determined in nuclear and cytoplasmic fractions. Treatment with 1 nM R1881 resulted in a significant increase of nuclear AR levels in all cell lines (Figure 3A,B). Although not statistically significant, nuclear AR levels in LNCaP cells were reduced after 10 μM Darolutamide treatment (Figure 3C). All other tested antiandrogens significantly reduced nuclear AR levels at a concentration of 10 μM, which was comparable to complete steroid hormone withdrawal (~0.15 fold of 1 nM R1881).

In C4-2 cells (Figure 3D), the presence of antiandrogens also resulted in a significant reduction in nuclear AR levels at a concentration of 10 µM, comparable to complete steroid hormone withdrawal (~0.30 fold of 1 nM R1881). At a concentration of 1 µM, Enzalutamide was already able to reduce nuclear AR levels to a similar extent.

In LAPC4 cells (Figure 3E), all antiandrogens significantly decreased nuclear AR levels at a concentration of 10 µM to a similar extent as complete steroid hormone withdrawal (~0.45 fold of 1 nM R1881). At a concentration of 1 µM, Bicalutamide and Apalutamide were already able to reduce nuclear AR levels to a similar extent.

Conversely, no significant cytoplasmic AR was observed after antiandrogen treatment (Appendix A).

### 3.4. Influence of Antiandrogen Treatment on AR mRNA and Protein Level

To measure the impact of antiandrogen incubation on *AR* mRNA levels, LNCaP, C4-2, and LAPC4 cells were starved for 24 h followed by treatment with 1 nM R1881 and different concentrations of antiandrogens for 16 h. None of the antiandrogens mediated any change in *AR* mRNA levels in the tested cell lines (Figure 4A–C).

To assess the influence of antiandrogens on AR and PSA protein levels, the cell lines were starved for 24 h followed by treatment with non-treatment, 1 nM R1881 treatment, and different concentrations of antiandrogens for 72 h. To determine changes in protein levels, western blot analyses on AR, PSA, and GAPDH were performed (Figure 5A–D). Changes in PSA levels were used as antiandrogen treatment control. All cell lines demonstrated decreased PSA levels after antiandrogen treatment, confirming antiandrogen response (Appendix A).

In LNCaP cells, besides Apalutamide, all antiandrogen treatments decreased AR protein levels to ~0.50 fold at 10 µM (Figure 5E). Although not statistically significant, only Enzalutamide and Apalutamide influenced AR protein levels in C4-2 cells (Figure 5F). In LAPC4 cells, all antiandrogens used in this study resulted in a significant concentration-dependent reduction in AR protein levels down to ~0.30 fold (Figure 5G).

When comparing the influence of the tested antiandrogens on AR protein stability, the results revealed that the cell line C4-2 responded with the lowest and LAPC4 with the highest sensitivity to antiandrogen treatment regarding AR protein levels (Figure 5H–K).

### 3.5. AR Protein Degradation Is an Early Event after Antiandrogen Treatment

As changes in AR protein stability have been identified as an early event after Enzalutamide treatment, the influence of Bicalutamide, Enzalutamide, Apalutamide, and Darolutamide on AR stability was tested in shorter periods [25]. To assess the impact of antiandrogens on AR protein levels, LAPC4 cells were starved for 24 h followed by treatment with non-treatment, 1 nM R1881 treatment, and 10 µM of antiandrogens for 2, 4, 6, 24, 48, and 72 h. To determine changes in protein levels, western blot analyses on AR and GAPDH were performed (Figure 6A,B). Densiometric analyses revealed that all tested androgens induce an early AR protein reduction (Figure 6B). Apalutamide caused the first significant effects after 2 h, and Bicalutamide and Apalutamide after 4 h. Darolutamide showed the first significant effects on AR protein levels after 6 h. In contrast, reduction of the AR target genes *PSA*, *TMPRSS2*, and *PSA* are less prominent after 2 h and 4 h (Figure 6C). The first significant results on AR target genes were seen after 6 h of Bicalutamide treatment (Figure 6C).

## 4. Discussion

AR signaling plays an essential role in prostate development and progression of PC and thus is a critical target in PC therapy [3,31,32]. For metastatic PC, the antiandrogens Bicalutamide, Enzalutamide, Apalutamide, and Darolutamide have shown promising therapy results [33,34,35,36,37]. However, even though these antiandrogens have been clinically investigated in highly controlled and consistent studies, their mechanism of action has been studied inconsistently. Therefore, this study aimed to compare the effects of the antiandrogens Bicalutamide, Enzalutamide, Apalutamide, and Darolutamide on AR-mediated gene transactivation, AR localization, AR expression, and cell viability, in vitro systematically. For this purpose, the cell lines LNCaP, LAPC4, and C4-2 were used, representing different stages of the disease. LNCaP and LAPC4 are hormone-sensitive PC cell lines, whereas the LNCaP sub-cell line C4-2 represents CRPC [17,19,20]. Moreover, LAPC4 expresses a wild-type AR, whereas LNCaP and C4-2 express mutated AR [8].

As the AR is the primary target of antiandrogens, they directly regulate the transcriptional activity of the receptor. This modulation of AR functionality is influenced by several factors such as the formation of heat shock protein (HSP) complexes, receptor dimerization, nuclear translocation, DNA binding, reduction of coactivator binding, and promotion of corepressor interaction [4,38]. The data presented here reveals that all tested antiandrogens reduce AR functionality in a concentration-dependent manner, confirmed by previous studies reporting AR inhibition by antiandrogens [13,16,39]. However, a 5–10 fold increased inhibitory effect of Enzalutamide, Apalutamide, and Darolutamide compared to Bicalutamide has been reported [13,16,39]. In this study, there was no significant advantage of any antiandrogen regarding the inhibition of AR functionality. The data revealed that cell line-specific characteristics had more effect on antiandrogens’ efficiency than the chemical compound itself. For example, Bicalutamide is the most potent antiandrogen in LNCaP cells and C4-2 cells, whereas all antiandrogens are similarly efficient in LAPC4 cells. In general, LAPC4 cells display the highest sensitivity, and C4-2 cells the lowest sensitivity to antiandrogen treatment of the tested cell lines. The comparable effects also reflect the influence of the tested antiandrogens on AR-mediated gene transactivation on cell viability. This result suggests that the cellular background of the cell lines has a significant impact on the molecular and cellular mode of action of the antiandrogens.

Among other mechanisms, sensitivity to antiandrogens is mediated by AR regulation and genetic changes, including AR overexpression, AR amplification, and AR somatic point mutations [32]. Point mutations within the AR gene have been shown to confer antiandrogen resistance, change ligand specificity, and receptor transcriptional activity [40]. The AR mutation F877L/F876L has been identified in Enzalutamide and Apalutamide treated patients. Enzalutamide and Apalutamide binding to the F877L mutated AR protein alter the antagonistic effects of the antiandrogens to agonistic effects [32,40,41,42]. Other AR mutations associated with low antiandrogen response are L702H, W742C/L, and T878A [40,42]. The hormone-sensitive cell line LNCaP and its castration-resistant sub-cell line C4-2 have harbored the T878A mutation (published initially as T868A) in the ligand-binding domain of the AR, whereas LAPC4 cells express the wildtype AR [8,20,43]. Thus, the T877A mutation in the AR may explain the diminished sensitivity of LNCaP and C4-2 cells to antiandrogen efficacy. Besides, in contrast to the LAPC4 cells and LNCaP cells, C4-2 cells express the ligand-independent AR splice variant V7 protein reported to mediate antiandrogen resistance [44,45]. Therefore, AR splice variants could also lead to the low antiandrogen response of C4-2 cells, as demonstrated in this study. However, other AR-associated factors such as AKT, STAT3, p300, and glucocorticoid receptor cannot be excluded in these antiandrogen resistance mechanisms [27,32,46].

Inhibition of AR translocation to the nucleus by antiandrogens is essential to prevent DNA binding and coactivator recruitment of the ligand-receptor complex and thus the transcriptional activity of the AR [3]. In contrast to second-generation antiandrogens, Bicalutamide has been reported not to inhibit AR nuclear translocation in the presence of R1881 [12,13,47]. However, the second-generation antiandrogens Enzalutamide, Apalutamide, and Darolutamide have been shown to suppress this nuclear translocation of the AR [3]. The data presented in this study revealed reduced nuclear AR protein after treatment with all tested antiandrogens at 10 µM. This result was not expected, as previous studies excluded the inhibition of the AR translocation into the nucleus by Bicalutamide [3]. However, the data from Clegg et al. already indicated a role of Bicalutamide in nuclear translocation, but less compared to Apalutamide or Enzalutamide [13]. Thus, changes in nuclear translocation should alter AR protein levels in both nuclear and cytoplasmic departments. Here, only a change in nuclear protein level after antiandrogen treatment could be observed, indicating a possible role of AR protein turnover in the regulation of nuclear AR. This hypothesis is in line with the observations by Lv et al. demonstrating that targeting the AR androgens cause MDM2 mediated nuclear AR degradation and not nuclear AR export [48]. Other studies with Enzalutamide, Apalutamide, and Darolutamide also confirmed reduced nuclear protein than in AR translocation to the cytoplasm [13,39,49]. This data is strengthened by the results of this study showing no increase in cytoplasmic AR levels after antiandrogen treatment compared to R1881 treated controls. Altogether, these studies and the data presented here suggest a pivotal role of nuclear AR protein stability in the mode of action of the tested antiandrogens.

AR protein levels are influenced by antiandrogen treatment [50]. In accordance with a previous study, the unaltered AR mRNA levels suggest that antiandrogens do not affect AR expression, but rather AR protein turnover, as AR stability [25]. Several mechanisms of AR protein regulation after androgen withdrawal and antiandrogen treatment have been described and linked to therapy resistance [51]. High levels of PIAS1 and STAT5, for example, enhance AR protein stability and therefore impair drug efficiency in the presence of Abiraterone and Enzalutamide [50,52]. Moreover, HSPs have been reported to stabilize AR protein and thus affect the antiandrogen’s inhibitory effects on the AR functionality [53]. High AR protein stability has also been linked to the castration-resistant cell line C4-2 after androgen withdrawal [54]. Here, treatment with all examined antiandrogens resulted in cell line-specific AR protein degradation response. C4-2 responded with the lowest sensitivity and LAPC4 with the highest sensitivity to antiandrogen treatment. The similarity to the AR functionality and cell viability results suggests a possible role of AR protein stability in the cellular response of PC to antiandrogens. Previously, a role of proteolytic degradation of the AR protein after Enzalutamide treatment has been identified in the cell line LNCaP [25]. The data revealed an Enzalutamide treatment-induced reduction in AR protein levels after 4 h accompanied by decreased cell proliferation. In addition, this study showed reduced AR protein levels in less than 6 h after treatment with all tested antiandrogens. Since this reduction of AR protein appears to occur before any modulation in AR functionality, it can be concluded that the reduction of AR protein levels has an early and substantial role in the mode of action of antiandrogens. This conclusion may also explain the promising results of AR degrading agents in CRPC and Enzalutamide resistant cell line models [5,54]. The precise mechanism of how antiandrogens regulate AR protein levels appears complex, and data on this is still limited. A better understanding of this mechanism may help to develop novel therapeutic strategies to increase antiandrogen efficiency, prevent therapy resistance, or re-sensitize antiandrogen resistant PC to antiandrogens. Since this understanding is of immense clinical relevance, further investigations should be conducted.

## 5. Conclusions

This study compared the AR inhibitory efficacy of the antiandrogens Bicalutamide, Enzalutamide, Apalutamide, and Darolutamide. The data demonstrated that all tested antiandrogens inhibit AR signaling to a similar extent. Significant differences were not found in comparing antiandrogens but were evident in cell line-specific differences in cell responses. Therefore, the cellular background, especially AR mutations, may represent an essential factor in antiandrogen efficiency. In addition, this study was able to show a correlation between the level of antiandrogen-induced AR protein reduction and the response of the cell lines to antiandrogen treatment and therefore identifies AR turnover as a possible indicator for antiandrogen treatment response.

## Figures and Tables

**Figure 1 life-11-00874-f001:**
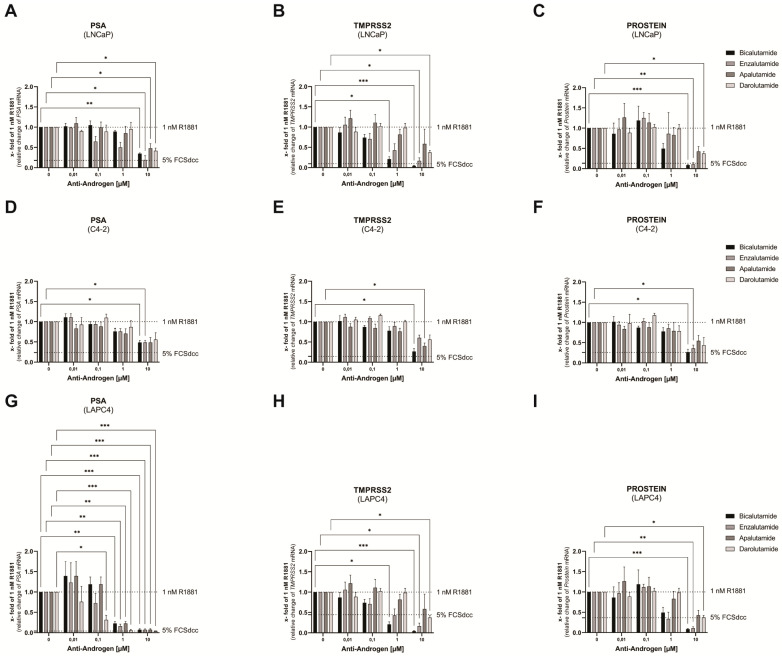
Influence of antiandrogens on the AR-mediated gene transactivation on androgen-dependent (LNCaP, LAPC4) and androgen-independent (C4-3) cell lines. To determine AR-mediated gene transactivation, modulation of mRNA levels of four AR target genes (*PSA, TMPRSS2, PROSTEIN*) were detected by qPCR and displayed as a mean change of 1 nM R1881. mRNA levels of AR target genes were normalized to the geometric mean of *TBP* and *HPRT1*. (**A**–**C**) Relative change of the AR target genes *PSA* (**A**), *TMPRSS2* (**B**), and *PROSTEIN* (**C**) after 16 h of 1nM R1881 and Bicalutamide, Enzalutamide, Apalutamide, or Darolutamide treatment in the cell lines LNCaP. (**D**–**F**) Relative change of the AR target genes *PSA* (**D**), *TMPRSS2* (**E**), and *PROSTEIN* (**F**) after 16 h of 1nM R1881 and Bicalutamide, Enzalutamide, Apalutamide, or Darolutamide treatment in the cell lines C4-2. (**G**–**I**) Relative change of the AR target genes *PSA* (**G**), *TMPRSS2* (**H**), and *PROSTEIN* (**I**) after 16 h of 1nM R1881 and Bicalutamide, Enzalutamide, Apalutamide, or Darolutamide treatment in the cell lines LAPC4. Values are expressed as mean ± SEM of at least three independent experiments. All differences highlighted by asterisks were statistically significant (*: *p* ≤ 0.05; **: *p* ≤ 0.01; *** *p* ≤ 0.001).

**Figure 2 life-11-00874-f002:**
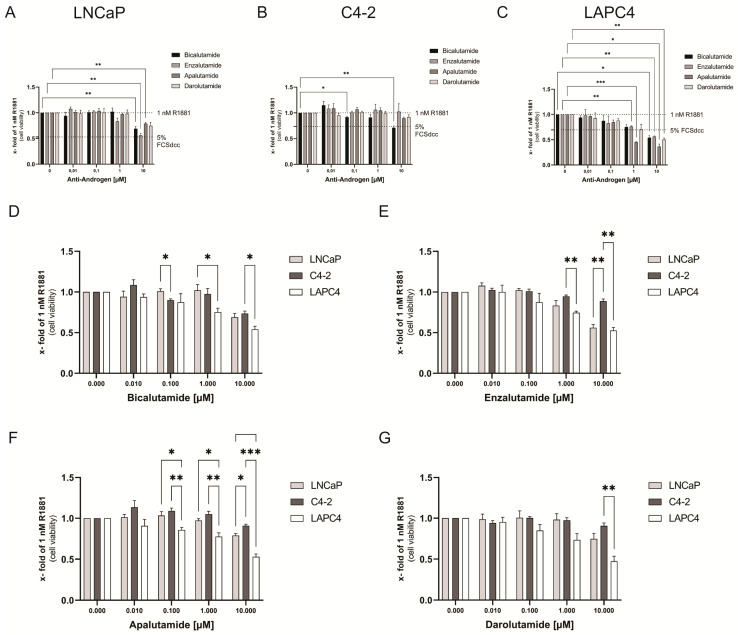
The cellular background is the primary driver of antiandrogen mediated effect on cell viability: (**A**–**C**) Relative change of cell viability after 72 h of 1 nM R1881 and Bicalutamide, Enzalutamide, Apalutamide, or Darolutamide treatment in the cell lines LNCaP (**A**), C4-2 (**B**), LAPC4 (**C**). The antiandrogen comparison revealed marginal differences between the used antiandrogens in the individual cell lines (**D**–**G**). Cell line response comparison between LNCaP, C4-2, and LAPC4 cells after treatment with 1 nM R1881 and Bicalutamide (**D**), Enzalutamide (**E**), Apalutamide (**F**), or Darolutamide (**G**). The cell line comparison revealed significant differences between the responses to antiandrogen treatment. Cell viability was assessed using the WST-1 cell proliferation reagent (Roche) 72 h after treatment. Values are presented as relative values of 1 nM R1881 treated cells and expressed as mean ± SEM of at least three independent replicates. All differences highlighted by asterisks were statistically significant (*: *p* ≤ 0.05; **: *p* ≤ 0.01; ***: *p* ≤ 0.001).

**Figure 3 life-11-00874-f003:**
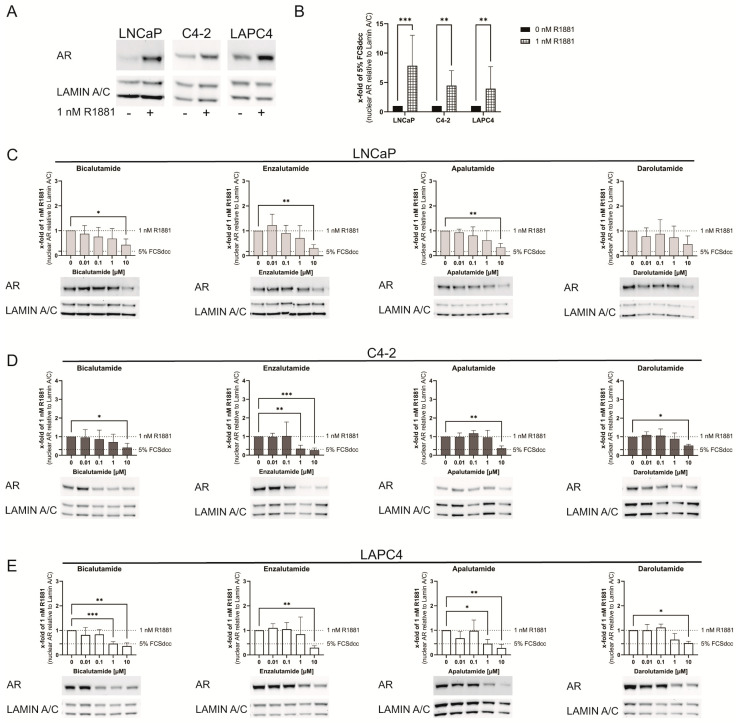
Antiandrogen treatment reduces nuclear AR protein levels. (**A**) Representative western blot of nuclear AR and Lamin A/C protein in LNCaP, C4-2, and LAPC4 after 1 nM R1881 treatment. All cell lines response with a significant increase of nuclear AR after R1881 treatment. (**B**) Relative change of nuclear AR protein after treatment with 1nM R1881 (**B**). (**C**–**E**) Relative change of nuclear AR protein after treatment with 1 nM R1881 and Bicalutamide, Enzalutamide, Apalutamide, or Darolutamide in LNCaP (**C**), C4-2 (**D**), and LAPC4 (**E**). All cell lines responded with a decrease of nuclear AR protein 2 h after antiandrogen treatment in a cell line-dependent manner. Lamin A/C (nuclear fraction) and GAPDH (cytoplasmic fraction) detection controlled fraction quality (Appendix A). Values are presented as relative values of 1 nM R1881 treated cells and expressed as mean ± SD of at least three independent experiments. All differences highlighted by asterisks were statistically significant (*: *p* ≤ 0.05; **: *p* ≤ 0.01; ***: *p* ≤ 0.001).

**Figure 4 life-11-00874-f004:**
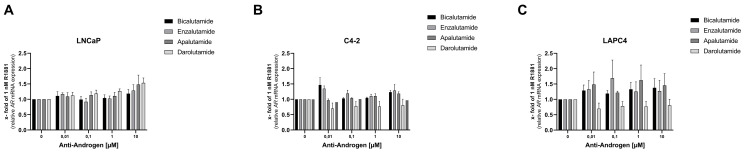
Antiandrogen treatment does not change *AR* mRNA level: Relative change of *AR* mRNA levels after treatment with 1 nM R1881 and Bicalutamide, Enzalutamide, Apalutamide, or Darolutamide in LNCaP (**A**), C4-2 (**B**), and LAPC4 (**C**). The antiandrogen treatment did not influence *AR* mRNA levels 16 h after treatment. Changes in mRNA levels have been assessed using qPCR. Values are presented as relative values of 1 nM R1881 treated cells and expressed as mean ± SEM of at least three independent experiments.

**Figure 5 life-11-00874-f005:**
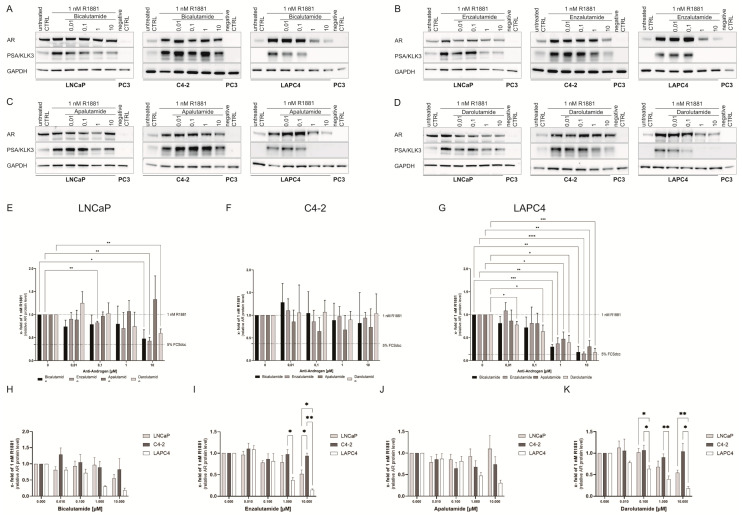
Influence of antiandrogen treatment on AR protein levels. (**A**–**D**) Representative western blots of AR protein, PSA, and GAPDH protein in LNCaP, C4-2, LAPC4, and PC3 after 1 nM R1881 and Bicalutamide (**A**), Enzalutamide (**B**), Apalutamide (**C**), or Darolutamide (**D**) treatment. PSA served as antiandrogen treatment control. PC3 cells were used as a negative control for PSA and AR. (**E**–**G**) Relative change of AR protein after treatment with 1 nM R1881 and Bicalutamide, Enzalutamide, Apalutamide, or Darolutamide in LNCaP (**E**), C4-2 (**F**), and LAPC4 (**G**). Cell line response comparison between LNCaP, C4-2, and LAPC4 cells of AR protein stability after treatment efficiency of Bicalutamide (**H**), Enzalutamide (**I**), Apalutamide (**J**), and Darolutamide (**K**). Changes in AR protein level have been assessed using western blot 72 h after treatment. Values are presented as relative values of 1 nM R1881 treated cells and expressed as mean ± SD of at least three independent experiments. All differences highlighted by asterisks were statistically significant (*: *p* ≤ 0.05; **: *p* ≤ 0.01; ***: *p* ≤ 0.001; ****: *p* ≤ 0.0001).

**Figure 6 life-11-00874-f006:**
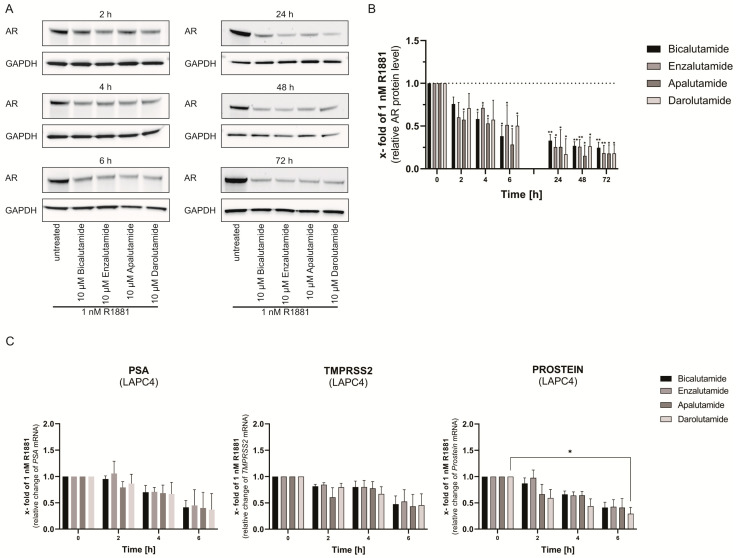
AR protein degradation is an early event after antiandrogen treatment. The analysis revealed that AR protein degradation starts before the reduction of AR-mediated gene transactivation activity. (**A**) Representative western blots of AR protein and GAPDH protein in LAPC4 after 2 h, 4 h, 6 h, 24 h, 48 h, and 72 h of 1 nM R1881 and 10 µM Bicalutamide, 10 µM Enzalutamide, 10 µM Apalutamide, or 10 µM Darolutamide treatment. (**B**) Relative change of AR protein after treatment with 1 nM R1881 and Bicalutamide, Enzalutamide, Apalutamide, or Darolutamide in LAPC4. Values are presented as relative values of 1 nM R1881 treated cells and expressed as mean ± SD of at least three independent experiments. (**C**) Relative change of AR-mediated gene transactivation activity after 2 h, 4 h, and 6 h treatment with 1 nM R1881 and 10 µM Bicalutamide, 10 µM Enzalutamide, 10 µM Apalutamide, or 10 µM Darolutamide treatment in LAPC4 cells. To determine AR-mediated gene transactivation, modulation of mRNA levels of four AR target genes (*PSA, TMPRSS2, PROSTEIN*) were detected by qPCR and displayed as a mean change of 1 nM R1881. Values are presented as relative values of 1 nM R1881 treated cells and expressed as mean ± SD (western blot) or mean ± SEM (qPCR) of at least three independent experiments. All differences highlighted by asterisks were statistically significant (*: *p* ≤ 0.05; **: *p* ≤ 0.01).

**Table 1 life-11-00874-t001:** Characteristics of the used cell lines.

Cell Line	Characteristics	Origin	Patients Background	Reference
LNCaP	AR (T877A), Androgen dependent	Lymph node	50-year-old Caucasian male	[19]
C4-2	AR (T877A), Androgen independent, LNCaP sub-cell line	Lymph node	50-year-old Caucasian male	[17]
LAPC4	AR wt, Androgen dependent	Lymph node	non applicable	[20]
PC3	Cells express no AR protein, Androgen independent, small cell neuroendocrine carcinoma	Bone metastasis	62-year-old Caucasian male	[21,22]

**Table 2 life-11-00874-t002:** Antibodies used in the study.

Name	Company	Lot	Dilution
Androgen Receptor (D6F11) XP Rabbit mAb	Cell Signaling Technology, Cambridge, UK	9	1:5000
Lamin A/C (4C11) Mouse mAb	Cell Signaling Technology, Cambridge, UK	5	1:2000
Mouse Monoclonal anti-GAPDH (6C5)	Novus Biologicals, Littleton, CO, USA	19/05-G4cc-C5cc	1:10,000
PSA/Kallikrein 3 (KLK3) (D6B1) XP Rabbit mAb	Cell Signaling Technology, Cambridge, UK	4	1:1000

## Data Availability

The data presented in this study are available in this article and Appendix A.

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
