# Peer review of "A Systematic Comparison of Antiandrogens Identifies Androgen Receptor Protein Stability as an Indicator for Treatment Response"

_life, 2021, doi:10.3390/life11090874_

Round 1

Reviewer 1 Report

The authors present an interesting study by comparing the effects of four antiandrogens in three different cell lines. Below are the lists of comments to improve the manuscript and the figures.

Major comments

  • Figure 1 shows AR transactivation activity inhibition by antiandrogens, displayed as a mean change of four different genes (PSA, TMPRSS2, PSMA, PROSTEIN). Basal expression of these genes in each cell lines may be quite different, which may affect antiandrogen sensitivity. For example, in Bicalutamide treated LNCaP cells, TMPRSS2, PROSTEIN and PSA are downregulated compared to non-treated cells, while PSMA is not changed. In 10 μM Enzalutamide treated LAPC4 cells, PSA is strongly repressed compared to control while PROSTEIN doesn’t change to such extent. In general PSMA expression does not show repression by antiandrogens. Rather, it increases with some antiandrogens. The authors state on page 4 lines 177-178 that “The lowest effect on the AR transactivation activity was shown in the presence of Enzalutamide, revealing no significant inhibition at all”. However, Figure S1 shows that PSA, TMPRSS2 and PROSTEIN are significantly repressed with Enzalutamide treatment in LNCaP cells. These observations prove that taking the average of these four genes is not an appropriate method to show AR transactivation activity. Figure 1 and Figure 6C should be modified to show the expression of the most representative AR target genes (PSA, TMPRSS2, PROSTEIN) independently.
  • Results that represent Figure 6C should be shown for each gene, as in Figure S1.
  • The sentence on page 4 line 193 “In general, all tested antiandrogens reduce AR activity to a similar extent in one cell line” needs to be reconsidered. For example, in LNCaP cells, Bicalutamide seems to show the most potent inhibition of AR transactivation activity after 10 nM R1881 treatment according to Figure 1A, differing from the other three antiandrogens. As the purpose of this study is a systematic comparison of different antiandrogens among cell lines of different background, if would be more helpful to readers that differences are highlighted.
  • In accordance to the comment above, the sentence on page 6 lines 227-228 “Similar to the AR-activity results, all tested antiandrogens reduce cell viability to a similar extent in a cell line-dependent manner” needs to be revised. It is not clear what “to a similar extent” means, as the effects of each antiandrogen is different in each cell line.

Minor comments

  • In Figure S1C there are two graphs showing PSMA expression in Bicalutamide treated cells, while Enzalutamide data is missing.
  • It would be helpful for the readers to show the differences and the characteristics about the three cell lines used in the study before the Results, in order to know what cellular background means. Thus, the sentences on page 10 lines 346-349 that explain the characteristics about LNCaP, LAPC4 and C4-2 can be shifted to the Introduction section.
  • The minimal concentration of Enzalutamide and Bicalutamide should be shown instead of 0, on page 2 lines 76-77.
  • The authors state on page 4 lines 175-177 that “Bicalutamide displays the most potent inhibition of AR transactivation activity after 1 nM R1881 treatment”. However, Figure 1A shows that 10uM Bicalutamide most strongly represses AR activity in LNCaP cells, which is not in accordance to the text.
  • The authors state on page 6 lines 222-223 that “Except for Enzalutamide, a decrease in cell viability starts at a concentration of 1 μM and peaks at 10 μM (Figure 2C)”. However, the figure seems that cell viability starts to decrease at 0.1 μM.
  • The authors state on pages 6-7 lines 247-248 “In LNCaP cells (Figure 3C), only Darolutamide could not significantly reduce nuclear AR levels“. However, the representative Western Blot clearly shows nuclear AR protein reduction in 10μM Darolutamide treated LNCaP cells. If the mean of the three independent experiments didn’t show statistical significance, sentences could be modified to the following sentences so that readers are not confused: “Although not statistically significant, reduction of nuclear AR levels in LNCaP cells was observed by 10 μM Darolutamide treatment. All other tested antiandrogens significantly reduced nuclear AR levels at a concentration of 10 μM to a similar extent of complete steroid hormone withdrawal (~0.15 fold of 1 nM R1881).
  • Figure 4 shows AR mRNA levels. The vertical axis (Y axis) and the legend are not correct.
  • Page 8 line 288: “In addition”, is not needed as the previous sentence already mentions PSA.
  • Page 8 line 290: (Figure S5S) should be corrected to (Figure S5D)
  • Figures 5E to 5K should be enlarged and presented in increased resolution, as letters cannot be read. Figures 3D to 3E and Figure 5A to 5D can also be improved.
  • The authors state on page 8 lines 292-293 that “In C4-2 cells, only Enzalutamide and 10 μM Apalutamide decreased AR protein levels (Figure 5F)”. However, 1 μM Apalutamide also seems to decrease AR protein levels.
  • Legend of Figure 5 (H-K) is incorrect. “cell viability” should be corrected to “protein stability”.
  • The authors state on page 9 lines 316-319 that “To assess the impact of antiandrogens on AR protein levels, the used cell lines were starved for 24 h followed by treatment with non-treatment, 1 nM R1881 treatment, and 10 μM of antiandrogens for 2, 4, 6, 24, 48, and 72 h.” However, Figure 6 only shows data using LAPC4 cells. The text or the figures need to be corrected.

Author Response

On the behalf of all authors, we would like to take this opportunity to express our sincere gratitude to the reviewers who identified areas of our manuscript that needed correction or modification. Their insightful comments have led to an improvement of our manuscript. Below you find the detailed response to the reviewers’ comments:

Major comments

  • Figure 1 shows AR transactivation activity inhibition by antiandrogens, displayed as a mean change of four different genes (PSA, TMPRSS2, PSMA, PROSTEIN). Basal expression of these genes in each cell lines may be quite different, which may affect antiandrogen sensitivity. For example, in Bicalutamide treated LNCaP cells, TMPRSS2, PROSTEIN and PSAare downregulated compared to non-treated cells, while PSMA is not changed. In 10 μM Enzalutamide treated LAPC4 cells, PSA is strongly repressed compared to control while PROSTEIN doesn’t change to such extent. In general PSMA expression does not show repression by antiandrogens. Rather, it increases with some antiandrogens. The authors state on page 4 lines 177-178 that “The lowest effect on the AR transactivation activity was shown in the presence of Enzalutamide, revealing no significant inhibition at all”. However, Figure S1 shows that PSA, TMPRSS2 and PROSTEIN are significantly repressed with Enzalutamide treatment in LNCaP cells. These observations prove that taking the average of these four genes is not an appropriate method to show AR transactivation activity. Figure 1 and Figure 6C should be modified to show the expression of the most representative AR target genes (PSA, TMPRSS2, PROSTEIN) independently.

We changed Figure 1 and the text according to the reviewers’ suggestions.

  • Results that represent Figure 6C should be shown for each gene, as in Figure S1.

As suggested, the results in Figure 6C have been changed

  • The sentence on page 4 line 193 “In general, all tested antiandrogens reduce AR activity to a similar extent in one cell line” needs to be reconsidered. For example, in LNCaP cells, Bicalutamide seems to show the most potent inhibition of AR transactivation activity after 10 nM R1881 treatment according to Figure 1A, differing from the other three antiandrogens. As the purpose of this study is a systematic comparison of different antiandrogens among cell lines of different background, if would be more helpful to readers that differences are highlighted

The paragraph about AR-mediated gene transactivation has been rewritten and the differences have been highlighted to improve clarity.

  • In accordance to the comment above, the sentence on page 6 lines 227-228 “Similar to the AR-activity results, all tested antiandrogens reduce cell viability to a similar extent in a cell line-dependent manner” needs to be revised. It is not clear what “to a similar extent” means, as the effects of each antiandrogen is different in each cell line.

To avoid generalization and conclusions in the results, the conclusion at the end of each result has been removed

Minor comments

  • In Figure S1C there are two graphs showing PSMA expression in Bicalutamide treated cells, while Enzalutamide data is missing.

As suggested by the reviewer the PSMA data has been removed from the manuscript. Therefore this issue is solved.

  • It would be helpful for the readers to show the differences and the characteristics about the three cell lines used in the study before the Results, in order to know what cellular background means. Thus, the sentences on page 10 lines 346-349 that explain the characteristics about LNCaP, LAPC4 and C4-2 can be shifted to the Introduction section.

A paragraph about the cell line models have been added to the introduction and a table with the main characteristics of the cell line models have been added to the material and method section.

  • The minimal concentration of Enzalutamide and Bicalutamide should be shown instead of 0, on page 2 lines 76-77.

The minimal concentration has been added

  • The authors state on page 4 lines 175-177 that “Bicalutamide displays the most potent inhibition of AR transactivation activity after 1 nM R1881 treatment”. However, Figure 1A shows that 10uM Bicalutamide most strongly represses AR activity in LNCaP cells, which is not in accordance to the text.

The paragraph about AR activity has been rewritten and the differences highlighted with greater clarity.

  • The authors state on page 6 lines 222-223 that “Except for Enzalutamide, a decrease in cell viability starts at a concentration of 1 μM and peaks at 10 μM (Figure 2C)”. However, the figure seems that cell viability starts to decrease at 0.1 μM.

The concentration has been corrected in the text

  • The authors state on pages 6-7 lines 247-248 “In LNCaP cells (Figure 3C), only Darolutamide could not significantly reduce nuclear AR levels“. However, the representative Western Blot clearly shows nuclear AR protein reduction in 10μM Darolutamide treated LNCaP cells. If the mean of the three independent experiments didn’t show statistical significance, sentences could be modified to the following sentences so that readers are not confused: “Although not statistically significant, reduction of nuclear AR levels in LNCaP cells was observed by 10 μM Darolutamide treatment. All other tested antiandrogens significantly reduced nuclear AR levels at a concentration of 10 μM to a similar extent of complete steroid hormone withdrawal (~0.15 fold of 1 nM R1881).

The paragraph has been changed as suggested by the reviewer

  • Figure 4 shows AR mRNA levels. The vertical axis (Y axis) and the legend are not correct.

The legend has been corrected

  • Page 8 line 288: “In addition”, is not needed as the previous sentence already mentions PSA.

“In addition” has been removed from the text.

  • Page 8 line 290: (Figure S5S) should be corrected to (Figure S5D)

The figure reference has been corrected

  • Figures 5E to 5K should be enlarged and presented in increased resolution, as letters cannot be read. Figures 3D to 3E and Figure 5A to 5D can also be improved.

The figure quality has been improved

  • The authors state on page 8 lines 292-293 that “In C4-2 cells, only Enzalutamide and 10 μM Apalutamide decreased AR protein levels (Figure 5F)”. However, 1 μM Apalutamide also seems to decrease AR protein levels.

The issue has been corrected in the manuscript

  • Legend of Figure 5 (H-K) is incorrect. “cell viability” should be corrected to “protein stability”.

The issue has been corrected in the manuscript

  • The authors state on page 9 lines 316-319 that “To assess the impact of antiandrogens on AR protein levels, the used cell lines were starved for 24 h followed by treatment with non-treatment, 1 nM R1881 treatment, and 10 μM of antiandrogens for 2, 4, 6, 24, 48, and 72 h.” However, Figure 6 only shows data using LAPC4 cells. The text or the figures need to be corrected.

The issue has been corrected in the manuscript

Reviewer 2 Report

  1. The manuscript could benefit from editing for grammar, missing words, and subject-verb agreement, etc. It is recommended that authors delete irrelevant "general" phrases and sentences, repeated and unneeded words. They should use short sentences. Also, some Introductory sentences are irrelevant or are not needed.
  2. All abbreviations should be revised and defined at their first use. For example, in the introduction, authors mentioned “the AR signaling blockade consists, i.a. of antiandrogen.” What is i.a.? Also, LHRH was not defined.
  3. Abstract: Authors did not elaborate on the cell lines background (which ones are androgen dependent and which are independent) in the methodology.
  4. The whole manuscript should be revised, references should be updated. For example, reference [1] should be updated and information from Siegel et al. 2021 could be used and cited instead.
  5. Introduction: The introduction is long, and details included could be moved to the discussion section.
  6. Introduction: Replace locally with locally in “and local advanced PC.”
  7. Introduction: Authors did not tackle all antiandrogens when listing them. For example, nilutamide and flutamide were not mentioned.
  8. Introduction: It is important also to elaborate in the introduction that several cell lines and PCa models (2D and 3D) are available, and those are divided into androgen dependent and androgen independent (PMID: 29477409, 27036046, 33195205).
  9. Introduction: The last paragraph of this section seems to be part of the abstract, where the main results are described. In my opinion, this paragraph should be replaced by the hypothesis and main objective of the study.
  10. Materials and Methods: Under cell culture, please add a table including the characteristics of the used cell lines and patient background from which those cell lines were primarily cultured.
  11. Materials and Methods: Add dilution factors for all antibodies used.
  12. Results: What is R1881? Please define and add more data about it and why it is used.
  13. Results: Remove the horizontal dotted lines from the bar graphs of Figures 1A, B, and C and 2A, B, and C. Also, same applies to other figures.
  14. Results: Are the treated concentrations of different drugs within the range of IC50 values? You must provide a reference for each IC50 value. In addition, why didn’t other drugs such as Docetaxel used on cell lines and treated simultaneously/in combination to assess the effect of adding a chemotherapy on AR signaling and the differential response on cell lines.
  15. Results: For all western blot images, please include densitometry readings/intensity ratio of each band. Also, add the MWs of proteins. In addition, please include the whole blot (uncropped blots) showing all the bands with all molecular weight markers on the Western in the Supplemental Materials?
  16. Figure legends can be revised as to be more informative of the results presented.
  17. In scientific writing, in general, symbols for genes are italicized whereas symbols for proteins are not italicized. The formatting of symbols for RNA and complementary DNA (cDNA) usually follows the same conventions as those for gene symbols. Gene names that are written out in full are not italicized (e.g., insulin-like growth factor 1). Genotype designations should be italicized, whereas phenotype designations should not be italicized.
  18. Discussion section is well written. Authors can focus more on the main findings and avoid repeating results presentation in the discussion. Authors could also correlate their findings with what has been published in literature. Clinical relevance should be added.

Author Response

On the behalf of all authors, we would like to take this opportunity to express our sincere gratitude to the reviewers who identified areas of our manuscript that needed correction or modification. Their insightful comments have led to an improvement of our manuscript. Below you find the detailed response to the reviewers’ comments:

  1. The manuscript could benefit from editing for grammar, missing words, and subject-verb agreement, etc. It is recommended that authors delete irrelevant "general" phrases and sentences, repeated and unneeded words. They should use short sentences. Also, some Introductory sentences are irrelevant or are not needed.

The authors have carefully reviewed the manuscript. In addition, the manuscript was checked thoroughly by a native speaker. As a result, grammar and sentence structure were improved by their  suggestions.

  1. All abbreviations should be revised and defined at their first use. For example, in the introduction, authors mentioned “the AR signaling blockade consists, i.a. of antiandrogen.” What is i.a.? Also, LHRH was not defined.

The manuscript has been thoroughly reviewed, and all abbreviations have now been introduced.

  1. Abstract: Authors did not elaborate on the cell lines background (which ones are androgen dependent and which are independent) in the methodology.

The information has been added to the abstract.

  1. The whole manuscript should be revised, references should be updated. For example, reference [1] should be updated and information from Siegel et al. 2021 could be used and cited instead.

The manuscript has been thoroughly reviewed, and several references have been updated.

  1. Introduction: The introduction is long, and details included could be moved to the discussion section.

As suggested, the introduction has been shortened.

  1. Introduction: Replace locally with locally in “and local advanced PC.”

Local has been replaced by locally.

  1. Introduction: Authors did not tackle all antiandrogens when listing them. For example, nilutamide and flutamide were not mentioned.

Information about the missing androgens have been added to the introduction

  1. Introduction: It is important also to elaborate in the introduction that several cell lines and PCa models (2D and 3D) are available, and those are divided into androgen dependent and androgen independent (PMID: 29477409, 27036046, 33195205).

Information about prostate cancer cell line models have been added to the introduction

  1. Introduction: The last paragraph of this section seems to be part of the abstract, where the main results are described. In my opinion, this paragraph should be replaced by the hypothesis and main objective of the study.

The authors carefully revised the manuscript on this issue. However, the abstract results part sum up the results of this manuscript, whereas the last paragraph leads to the problem of the non-comparability of the published data on the new antiandrogens. Thus, the last part prepares the aim of the study. Therefore, we did not replace this paragraph.

  1. Materials and Methods: Under cell culture, please add a table including the characteristics of the used cell lines and patient background from which those cell lines were primarily cultured.

The missing information has been added to the manuscript.

  1. Materials and Methods: Add dilution factors for all antibodies used.

The missing information has been added to the manuscript.

  1. Results: What is R1881? Please define and add more data about it and why it is used.

The missing information has been added to the materials and methods section of the manuscript.

  1. Results: Remove the horizontal dotted lines from the bar graphs of Figures 1A, B, and C and 2A, B, and C. Also, same applies to other figures

We want to thank the reviewer for their suggestion. The authors discussed this issue carefully. In our opinion, these dotted lines are essential as they represent the R1881 treated control and the level of absolute steroid depletion. As they represent two critical pieces of information, we decided not to change the figure.

  1. Results: Are the treated concentrations of different drugs within the range of IC50 values? You must provide a reference for each IC50 value. In addition, why didn’t other drugs such as Docetaxel used on cell lines and treated simultaneously/in combination to assess the effect of adding a chemotherapy on AR signaling and the differential response on cell lines.

We like to thank the reviewer for their  comment. As previous published IC50 are not comparable, this study is aimed to systematically compare the currently used antiandrogens Bicalutamide, Enzalutamide, Apalutamide, and Darolutamide, using identical experimental setups and drug concentrations. However, we added some published IC50 to material and methods to justify our treatment concentration range.

In this manuscript, we focus on drugs targeting the androgen receptor directly. As docetaxel's primary target is inhibition of microtubular depolymerization, it was not of interest for this study. However, as inhibition of microtubular depolymerization also affects AR translocation (PMID: 29900882) it would be interesting to invest in further studies.

  1. Results: For all western blot images, please include densitometry readings/intensity ratio of each band. Also, add the MWs of proteins. In addition, please include the whole blot (uncropped blots) showing all the bands with all molecular weight markers on the Western in the Supplemental Materials?

The data has been added to the supplemental materials

  1. Figure legends can be revised as to be more informative of the results presented.

We revised the figure legends and tried to add more information.

  1. In scientific writing, in general, symbols for genes are italicized whereas symbols for proteins are not italicized. The formatting of symbols for RNA and complementary DNA (cDNA) usually follows the same conventions as those for gene symbols. Gene names that are written out in full are not italicized (e.g., insulin-like growth factor 1). Genotype designations should be italicized, whereas phenotype designations should not be italicized.

We thank the reviewer for their  comment. We revised the manuscript according to their suggestion.

  1. Discussion section is well written. Authors can focus more on the main findings and avoid repeating results presentation in the discussion. Authors could also correlate their findings with what has been published in literature. Clinical relevance should be added.

We want to thank the reviewer for their  comment and added why this work has clinical relevance in our opinion.

Round 2

Reviewer 2 Report

Thank you for addressing all comments.